# Structure of human ORP3 ORD reveals conservation of a key function and ligand specificity in OSBP-related proteins

Junsen Tong[☯], Lingchen Tan[☯], Young Jun Im[☯]*

College of Pharmacy, Chonnam National University, Gwangju, Republic of Korea

☯ These authors contributed equally to this work.
* imyoungjun@jnu.ac.kr

## Abstract

Human ORP3 belongs to the oxysterol-binding protein (OSBP) family of lipid transfer proteins and is involved in lipid trafficking and cell signaling. ORP3 localizes to the ER-PM interfaces and is implicated in lipid transport and focal adhesion dynamics. Here, we report the 2.6–2.7 Å structures of the ORD (OSBP-related domain) of human ORP3 in apo-form and in complex with phosphatidylinositol 4-phosphate. The ORP3 ORD displays a helix grip β-barrel fold with a deep hydrophobic pocket which is conserved in the OSBP gene family. ORP3 binds PI(4)P by the residues around tunnel entrance and in the hydrophobic pocket, whereas it lacks sterol binding due to the narrow hydrophobic tunnel. The heterologous expression of the ORDs of human ORP3 or OSBP1 rescued the lethality of seven ORP (yeast *OSH1-OSH7*) knockout in yeast. In contrast, the PI(4)P-binding site mutant of ORP3 did not complement the *OSH* knockout cells. The N-terminal PH domain and FFAT motif of ORP3 are involved in protein targeting but are not essential in yeast complementation. This observation suggests that the essential function conserved in the ORPs of yeast and human is mediated by PI(4)P-binding of the ORD domain. This study suggests that the non-vesicular PI(4)P transport is a conserved function of all ORPs in eukaryotes.

## Introduction

The lipid compositions of membrane bilayers in eukaryotes are unique for each intracellular organelles and the proper distribution of lipids is crucial for cell function [1]. Various phosphoinositide lipids are specifically localized in the membranes of the subcellular organelles and serve as major regulators in membrane trafficking, and lipid signaling by re recruiting and/or activating effector proteins [2]. Non-vesicular lipid transport by soluble protein carriers plays a significant role in lipid distribution between the organelles that are not connected by membrane trafficking pathways [3].

Oxysterol-binding proteins (OSBP)-related proteins (ORPs) compose a large conserved family of lipid-binding proteins in eukaryotes and are implicated in many cellular processes including cell signaling, vesicular trafficking, lipid metabolism, and transport [4]. Mammals

**Data Availability Statement:** The coordinates and structure factors have been deposited in the Protein Data Bank with the accession codes, 7DEJ, and 7DEI for apo ORP3 ORD and the ORP3-PI(4)P complex, respectively.

**Funding:** This work was supported by the National Research Foundation of Korea (NRF) grant funded by the Ministry of Education, Science and Technology (grant no. 2019R1A2C1085530). The funders had no role in study design, data collection and analysis, decision to publish, or preparation of the manuscript.

**Competing interests:** The authors have declared that no competing interests exist.

have twelve ORP genes and their gene splicing increases the number of different protein products [5]. Human ORPs are grouped into six subfamilies depending on their sequence similarity and gene structure [5]. A common feature for all ORPs is the C-terminal OSBP-related ligand-binding domain (ORD) that binds phosphatidylinositol 4-phosphate, PI(4)P [6, 7]. The ORDs of many ORPs were known to bind other lipids such as sterols or phosphatidylserine (PS) in exchange of PI(4)P [6, 8, 9]. Most ORPs have targeting domains in the N-terminal region such as a pleckstrin homology (PH) domain that binds phosphoinositides, and an ER-targeting FFAT (two phenylalanines in an acidic tract) motif [10]. Some ORPs have a C-terminal transmembrane segment that anchors the proteins to the ER membrane or ankyrin repeats for protein-protein interactions.

Recent discoveries suggested that PI(4)P-coupled lipid transport is one of the major roles of ORPs [11]. Many members of the OSBP family bind PI(4)P as a primary ligand and perform vectorial transport of secondary lipids such as sterol or PS by dissipating a PI(4)P gradient [6, 11]. PI(4)P is a common ligand for all ORPs, but other phosphoinositides such as PI(3,4)$P_2$ and PI(4,5)$P_2$ have been found to be a ligand for ORP1, ORP2, ORP5, and ORP8 [12–14].

The long ORPs containing the N-terminal targeting modules seem to transport lipids efficiently at the membrane contacts sites because they diffuse only a short distance between the membranes. Besides the role as LTPs, diverse functions of ORPs have been reported as sterol regulators or sensors that transfer information to a spectrum of different cellular processes [4]. Human ORP3 localizes to the ER-PM contacts and recruits R-Ras, a small GTPase that controls cell adhesion and migration [15]. Localization of ORP3 to the ER-PM contact sites is regulated by $Ca^{2+}$ and PKC dependent phosphorylation, which controls focal adhesions dynamics [16–18]. ORP3 is known to extract PI(4)P but no PS, PI(4,5)$P_2$, DAG or sterols [16]. So far, it is not known clearly how the lipid transfer activity of ORP3 is coupled to these various cellular processes.

In this study, we determined the structures of human ORP3 ORD in apo-form and in complex with PI(4)P. We found that ORP3 lacks sterol-binding activity while PI(4)P-binding to its ORD is essential for function. The ORDs of human OSBP1 and ORP3 rescued the growth defect of the yeast strains with *OSH1-7* knockout, suggesting the essential role of ORDs in PI(4)P regulation is shared between eukaryotic species. The structural analysis of ORP3 is consistent with the role of ORP3 as a PI4P transporter or as a regulator at membrane contact sites.

## Materials and methods

### 1. Cloning of human ORP3 ORD

The full length clone for human ORP3 gene (hMU001671, UniProt id: Q9H4L5) was purchased from 21C Frontier Human Gene Bank (KRIBB, Daejeon, Republic of Korea). DNA encoding the OSBP-related domain (residues 504–887) of human ORP3 was amplified by polymerase chain reaction and subcloned into the EcoRI/XhoI site of a modified pGEX-4T vector. The ORP3 ORD was tagged with an N-terminal hexahistidine-glutathione S-transferase (GST) followed by a thrombin protease cleavage site (LVPR/GS). To prevent oxidation of exposed cysteine residues located upstream of the lid, we replace the two cysteine residues to serine (C515S and C520S). The dipeptide residues (Asp860-Asp861) in the loop β19-β20 which were predicted to be disordered based on the secondary structure prediction (Predict-Protein http://predictprotein.org) were mutated to a glycine residue to improve the crystallization properties by surface entropy reduction. *Escherichia coli* strain BL21(DE3) cells transformed with the plasmids encoding the ORP3 ORD were grown to an OD600 of 0.8 at 37˚C in LB medium. Cells were induced by the addition of isopropyl β-D-1-thiogalactopyranoside to a final concentration of 0.5 mM and were incubated for 12 h overnight at 20˚C prior to harvesting.

## 2. Protein expression and purification

Cells expressing ORP3 ORD were resuspended in 2X PBS buffer containing 20 mM imidazole (lysis buffer) and lysed by sonication. The supernatant containing His-GST-ORD was applied onto a Ni-NTA affinity column. The Ni-NTA column was thoroughly washed with the lysis buffer. The fusion protein was eluted from the column using a buffer containing 100 mM Tris–HCl pH 8.0 (final), 300 mM imidazole. The eluate was concentrated to 10 mg/ml using Amicon Ultra-15 centrifugal filter and the His-GST tag was removed by cleavage with 10 international unit (IU) of thrombin protease (Reyon Pharmaceutical) per 10 mg of recombinant protein. The ORP3 ORD (isoelectric point 7.2) was separated from His-GST by HiTrap SP cation-exchange chromatography (GE healthcare). The ion exchange chromatography was performed using a linear gradient with buffer A composed of 50 mM MES-NaOH pH 6.5 and B buffer composed of 50 mM MES-NaOH pH 6.5, 500 mM NaCl. The fractions containing the ORP3 ORD were subjected to size-exclusion chromatography on a Superdex 200 column equilibrated with 20 mM Tris–HCl pH 8.0, 150 mM NaCl. The fractions containing the ORP3 ORD were concentrated by the centrifugal filter to 10 mg/ml for crystallization.

## 3. Crystallization and crystallographic analysis

Preliminary crystallization experiments were carried out at 22°C in 96-well crystallization plates using customized crystallization screening solutions by dispensing 0.8 μl protein solution and 0.8 μl precipitant solution. To obtain the crystals of the ORP3 ORD complexed with a phosphoinositide, we incubated the purified protein with two times molar ratio of PI(4)P diC8. The crystals of ORP3 ORD appeared after 5 d using a solution consisting of 0.1 M HEPES-NaOH pH 7.0, 25% PEG 1500, 0.2 M $MgCl_2$. The crystallization condition was further optimized to 0.1 M MES-NaOH pH 6.0, 25% PEG 1500, 0.1 M $MgCl_2$ using the hanging-drop technique in 15-well screw-cap plates. A drop consisting of 2 μl protein solution was mixed with 2 μl precipitant solution and equilibrated against 1 ml reservoir solution. High-quality crystals with dimensions of 0.1 X 0.1 X 0.15 mm appeared in one week. Crystals of the ORP3 ORD were cryoprotected in reservoir solution supplemented with 10% glycerol and flash-cooled by immersion in liquid nitrogen. Crystals were preserved in a cryogenic $N_2$-gas stream (~100 K) during diffraction experiments. Native diffraction data for ORP3 ORD were collected at a fixed wavelength of 0.97949 Å using an ADSC Q270 CCD detector on the 7A beamline at Pohang Light Source (PLS), Pohang Accelerator Laboratory. All data were processed and scaled using HKL-2000. The structure of ORP3 ORD was determined by molecular replacement using the structure of yeast Osh3 ORD (PDB code: 4INQ) as a search model. Two molecules of ORDs were found in the asymmetric unit using the program Phaser [19] and the density-modified map showed clear electron densities of the ORP3 ORDs. The final model including two molecules of ORP3 ORD bound with PI(4)P was refined to $R_{work}$/$R_{free}$ values of 20.0%/26.9% using the software Phenix [20].

## 4. DHE binding assay

DOPC (1,2-dioleoyl-*sn*-glycero-3-phosphocholine), DOPE (1,2-dioleoyl-*sn*-glycero-3-phosphoethanolamine), POPS (1-palmitoyl-2-oleoyl-*sn*-glycero-3-phospho-l-serine), were obtained from Avanti Polar Lipids, Inc. Phosphatidylinositol (soybean) and DHE were obtained from Sigma-Aldrich. To prepare DHE loaded liposomes, lipids dissolved in chloroform or in ethanol were mixed at the desired molar ratio, incubated at 37°C for 5 min and the solvent was evaporated by nitrogen stream. The dried lipids were resuspended in 50 mM HEPES-NaOH pH 7.2, and 120 mM potassium acetate (HK buffer) by vortexing. The liposomes were prepared at a total lipid concentration of 1.3 mM. The hydrated lipid mixture was

frozen and thawed five times using water bath and cooled ethanol at -70˚C. Liposomes were generated by extruding the lipid mixture 10 times through a 0.1 μm (pore size) polycarbonate filter. Fluorescent measurements of DHE binding to the ORDs was based on fluorescence resonance energy transfer (FRET) between Trp and bound DHE and were carried out using the methods previously reported [21]. The 1 ml of sample initially contained liposomes (1.3 mM) was diluted 2 times with HK buffer. The 2 ml of liposomes was placed in a square quartz cell, continuously stirred with a small magnetic bar at ambient temperature. Osh4 or ORP3 ORD proteins were injected to the sample cell with a final protein concentration of 0.5 mM and incubated for 30 min. Emission spectra were recorded upon excitation at 285 nm using a spectrofluorometer (FP-6200; JASCO).

## 5. Yeast complementation assay

DNAs coding for the various constructs of ORP3 and OSBP1 were cloned to pRS416MET25-GFP vector. Mutant constructs were generated by site-directed mutagenesis using the protocols provided by QuickChange Site-Directed Mutagenesis kit (Agilent Technologies) and confirmed by DNA sequencing. Complementation analysis was performed by testing the rescues from the growth defects of the temperature-sensitive yeast strain, CBY926 (*oshΔ*[*CEN*, osh4$^{ts}$]) by introduction of the ORP alleles [22]. The plasmids encoding the ORP genes were transformed to the CBY926 strain. As a control, the plasmids were transformed to the wild type strain, CBY1 (SEY6210, *MATα leu2-3112 ura3-52 his3Δ200 lys2-801 trp1Δ901 suc2Δ9*). The CBY926 yeast cells were grown in a Hartwell's complete (HC) medium without uracil at a permissive temperature (30˚C), and the five-fold dilution series were spotted on a HC medium agar plate and further incubated for 24h at 37˚C. The CBY1 cells were grown at 30˚C for all procedures.

## 6. Western blot

The CBY926 cells harboring GFP-ORP3 ORD wt and GFP-ORP3 ORD K603E plasmids were grown to the OD600 of 1.5. The cells were resuspended in 0.1 M NaOH and incubated for 5 min at room temperature. The isolated cells were resuspended in SDS-PAGE sample buffer and were boiled for 3 min. The supernatants were collected and loaded to SDS-PAGE. For the Western blot, the proteins in the SDS-PAGE gel were transferred to a PVDF membrane and the transfer was confirmed by Ponceau S staining. GFP monoclonal mouse IgG (Thermo-Fisher, #MA5-15256) was used for a primary antibody with a dilution ratio of 1:3000. The HRP-coupled rabbit anti-mouse IgG was used for the secondary antibody with a dilution ratio of 1:5000. The chromogenic detection of horseradish peroxidase activity of the secondary antibody was done using CN/DAB substrate and the blot was imaged by Azure c300 gel imaging system. The expression levels of the GFP-ORDs were analyzed by quantifying the band intensities using ImageJ software.

## 7. Microscopy

Yeast CBY926 cells were transformed with the pRS416MET25 vector containing GFP-ORP3 alleles. Yeast cells expressing the appropriate alleles were grown to an OD600 of 0.4–0.6 in a HC medium. The yeast cells were grown at a permissive temperature, 30˚C to allow growth of the yeast cells harboring the non-complementing ORP3 constructs. The cells were harvested and washed two times with PBS and resuspended in the selection medium for observation under the microscope. Visualization of cells was performed on an Eclipse Ti fluorescence microscope (Nikon) equipped with fluorescein isothiocyanate (FITC) filter and the images were captured with an Andor ixon EMCCD camera. The green fluorescence of GFP-ORP3

constructs were imaged with 8 sec exposure. The images of yeast cells were cropped using the software ImageJ [23].

## Results

### Structure of human ORP3 ORD

Human ORP3 contains a PH domain (residues 51–148) in the N-terminal region and a C-terminal ORD domain (residues 504–887). The middle region between the PH domain and ORD contains a predicted helical region (residues 330–401), and an FFAT motif (EFF-DAQE, residues 450–456) (Fig 1A). To obtain structural insight into the ligand recognition of the ORP3 ORD, we performed x-ray crystallographic studies of the ORD domain. We obtained the crystals of the ORP3 ORD in apo-form and in complex with PI(4)P diC8 with P3$_1$21 space group. In the asymmetric unit of the crystals, there were two molecules of ORP3 ORD related by two-fold non-crystallographic symmetry. The recombinant protein of the human ORP3 ORD was monomeric when analyzed by size exclusion chromatography. The dimer in the asymmetric unit seemed to form by lattice interaction during crystallization process. The structures of the ORP3 ORDs in apo- and PI(4)P-bound forms were determined at 2.6–2.7 Å resolution by the molecular replacement method using the structure of yeast Osh3 ORD (Table 1). The electron densities of PI(4)P ligand were clearly visible (Fig 1B). The ORP3 ORD has a central ligand-binding tunnel with a flexible lid covering the tunnel entrance (Fig 1C). The ORD core is made of a central antiparallel β-sheet of 20 strands that forms an incomplete β-barrel (Fig 1C and 1D). The hydrophobic tunnel in the center of the barrel has a cavity volume of 1600 Å$^3$. The tunnel is flanked by an N-terminal subdomain (residues 550–609) and a large C-terminal subdomain (residues 773–887). The N-terminal subdomain located inside the barrel consists of a two-stranded β-sheet and four α-helices that form a 35 Å-long antiparallel bundle. The C-terminal subdomain following the barrel contains four α-helices and two β-strands forming a hairpin. The residues 525–549 form a lid covering the tunnel opening. The lid of ORP3 ORD contains two turns of an amphipathic α-helix. The residues upstream of the lid (residues 510–524) form an extended loop on the concave surface of the C-terminal region (Fig 2A). The loop residues 521–524, and the loop at the C-terminal region (residues 874–881) were disordered in the structure with the high B-factors in the structure (Fig 2B).

The structures of the ORP3 ORDs in apo-form or in PI(4)P complex have a closed conformation of the lid by the hydrophobic interactions between the lid and the entrance of the hydrophobic tunnel. The backbone atoms of the residues 541–545 in the lid have hydrophilic contacts with the polar head group of PI(4)P. Three arginine residues (511–513) upstream of the lid make electrostatic interaction with the surface of the C-terminal subdomain (Fig 2A). Two conserved arginine residues (residues 512–513) make salt bridges with the acidic loop β3-β4 and Glu794 in the groove formed by the β17-α6 and α6-α7 loops. Leu516 is accommodated in the hydrophobic pocket formed by helix α6 and the loop β17-α6. These interactions seem to stabilize the N-terminal base of the lid favoring a closed conformation of the lid. In contrast, the lid of Osh4 lacking the upstream residues is completely open and disordered when ligand is absent [24]. The N-terminal upstream residues of the lid with a consensus sequence of (R/K/H)-R-X-X-(L/I)-P are conserved in OSBP1, ORP2, ORP3, ORP4, ORP6, and ORP7. The human ORP3 ORD has 35% sequence identity (52% similarity) to yeast Osh3 ORD over 377 residues and shows a high degree of structural conservation with the C$_\alpha$ rmsd of 1.3 Å for 311 equivalent residues (Fig 2C). The structure of ORP3 in this study is almost identical to the structure of human ORP3 previously reported [25] with the C$_\alpha$ rmsd of 0.437 Å (Fig 2D).

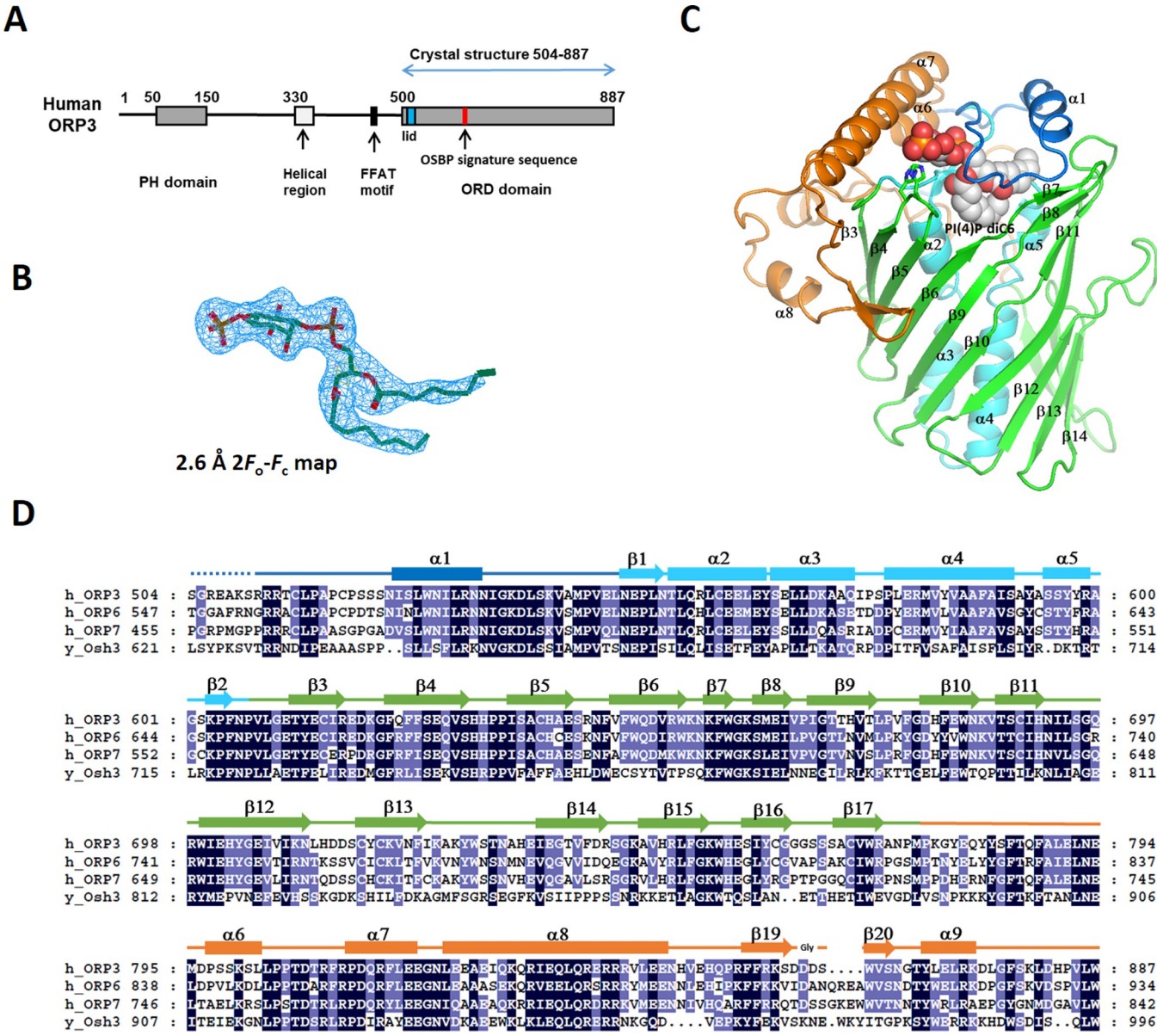

**Fig 1. Structure of the ORP3 ORD.** (A) Schematic representation of the domain structures of human ORP3. (B) 2.6 Å $2F_o-F_c$ electron density map of the PI(4)P diC6 contoured at 1σ. (C) The overall structure of ORP3 ORD. The N-terminal lid (residues 511–547) is shown in blue, the central helices (548–608) in cyan, the β-barrel (609–772) in green and the C-terminal subdomain (773–886) in orange. Bound PI(4)P diC6 is shown in a sphere model. (D) Sequence alignments of the ORD domains of human ORPs and yeast Osh3.

## Binding of PI(4)P to the ORP3 ORD

The ORP3 ORD was known to bind PI(4)P with a $K_d$ of 29 nM [25]. However, ORP3 has no binding affinity to PS, PI(4,5)$P_2$, and sterols [16]. This structural study demonstrates the key determinants of the PI(4)P recognition by the ORP3 ORD. Structural comparison shows that the closed form of apo ORD has a very similar conformation to the PI(4)P-bound form with a $C_\alpha$ rmsd of 0.49 Å. The ligand dependent conformational change is limited to the residues that recognize the PI(4)P head group around the tunnel entrance (Fig 3E). The PI(4)P molecule was well ordered in the crystal structure with clear electron

**Table 1. Data-collection and refinement statistics.**

| Crystal | Apo ORP3 ORD | ORP3 ORD—PI(4)P diC8 complex |
|---|---|---|
| Data collection | | |
| Beamline | PLS-5C | PLS-7A |
| Wavelength (Å) | 0.97934 | 0.97950 |
| Space group | $P3_121$ | $P3_121$ |
| Unit-cell parameters (Å, °) | $a$ = 95.9 Å, $b$ = 95.9 Å, $c$ = 188.9 Å | $a$ = 95.8 Å, $b$ = 95.8 Å, $c$ = 190.5 Å |
| Rotation range (Å) | 50–2.7 (2.75–2.70) | 50–2.6 (2.64–2.40) |
| No. of reflections | 127073 | 196407 |
| No. of unique reflections | 28026 (1409) | 31618 (1364) |
| Multiplicity | 4.5 (4.8) | 6.2 (6.2) |
| Mean $I/\sigma(I)$ | 35.1 (6.1) | 39.0 (4.4) |
| Completeness (%) | 98.8 (100) | 99.0 (88.3) |
| $R_{merge}$ (%) | 7.6 (36.8) | 12.7 (53.0) |
| Refinement | | |
| Resolution range (Å) | 50–2.7 (2.80–2.70) | 50–2.6 (2.67–2.60) |
| $R_{work}$ (%) | 23.3 (30.4) | 20.0 (31.9) |
| $R_{free}$ (%) | 28.5 (41.4) | 26.9 (43.9) |
| R.m.s.d., bond lengths (Å) | 0.012 | 0.011 |
| R.m.s.d., bond angles (°) | 1.330 | 1.612 |
| $B$ factor (Å$^2$) | | |
| Overall | 71.0 | 61.1 |
| Molecule A | 88.8 | 66.9 |
| Molecule B | 53.5 | 55.6 |
| Ligands | - | 62.2 |
| Water | 54.0 | 52.0 |
| No. of non-H atoms | | |
| Protein | 6155 | 6172 |
| Ligand | 0 | 86 |
| Solvent | 58 | 55 |
| Ramachandran statistics | | |
| Favored (%) | 92.8 | 93.5 |
| Outliers (%) | 0.13 | 0.67 |
| PDB entry | 7DEJ | 7DEI |

densities of the ligands. PI(4)P binds to the ORP3 ORD by inserting its two acyl chains into the hydrophobic tunnel. The phospho-inositol group of PI(4)P is recognized by the shallow basic pocket in the tunnel entrance. The conserved residues Lys829, Glu833, Arg836, and Arg837 from helix α8 make direct interaction with the phospho-inositol group (Fig 3A). The polar inositol head group makes many direct hydrophilic interactions with the protein, accounting for the specific recognition of PI(4)P by the ORP3 ORD. The 4-phosphate group makes hydrogen-bonds with the side-chains of the conserved His631 (β4-β5), His632 (β4-β5) and Arg837 (α8). The 1-phosphate group connecting the inositol ring and the glycerol moiety is hydrogen bonded to the conserved Lys603 (β2), Lys829 (α8) and the backbone amide atom of Met660 in the lid (Fig 3A). The 5- and 6- hydroxyl groups of the inositol ring make direct hydrogen bonds with Gln836 (α8) and Glu833, respectively. The ORP3 ORD is unable to bind $PI(4,5)P_2$ as the 5-phosphate group causes a steric clash with the side chains of the residues in the helix α8. Binding of $PI(3,4)P_2$ also seems unfavorable due to the clashes of the 3-phosphate group with the lid in a closed conformation. The structural

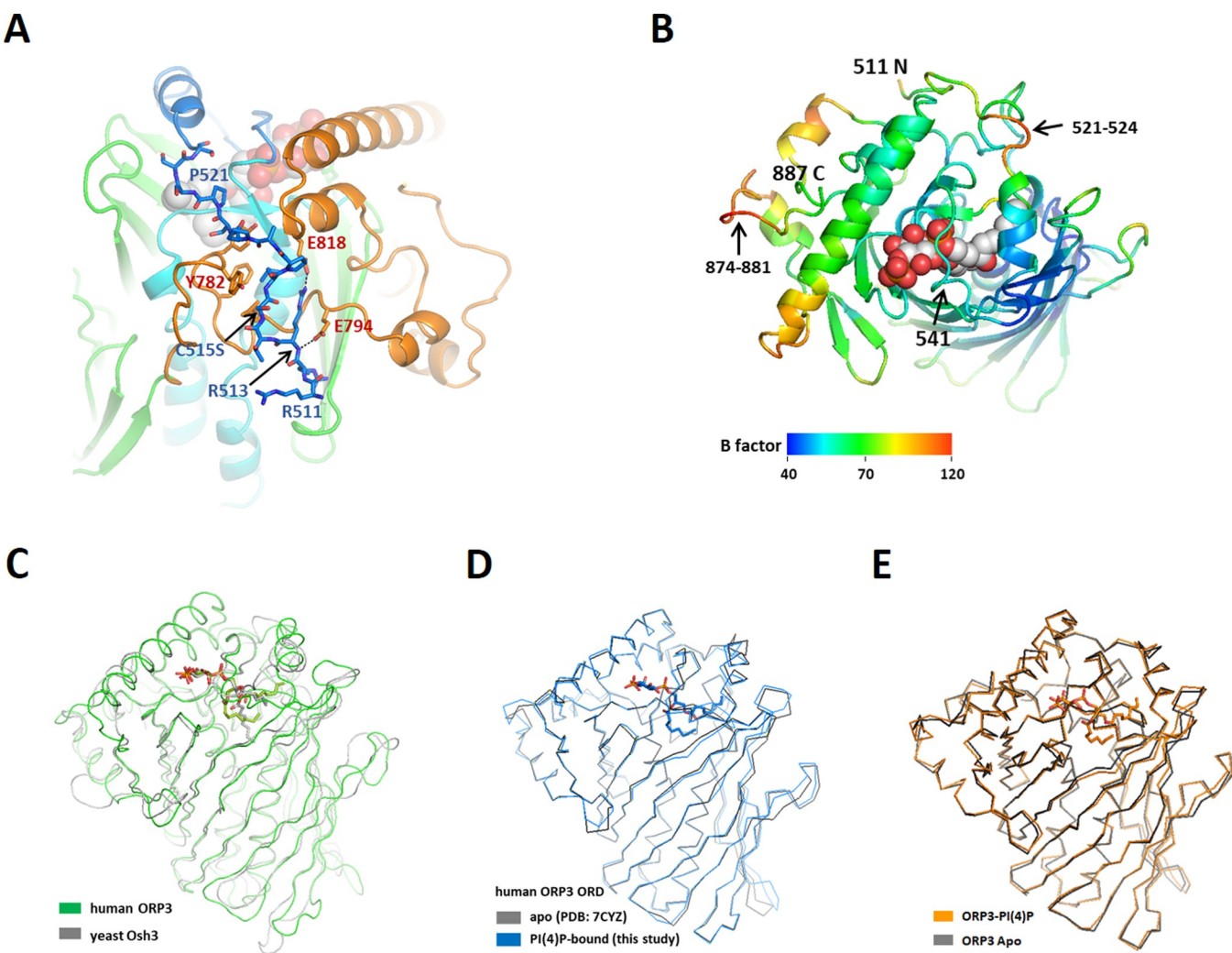

**Fig 2. Structure comparison of ORDs.** (A) The N-terminal upstream residues of the lid. The residues 511–524 are shown in stick models. (B) B-factor representation of the ORP3 ORD. Top view of Fig 2A. (C) Structure superposition of the ORD domains from human ORP3 and yeast Osh3. The PI(4)P of ORP3 is shown in cyan and the PI(4)P of yeast Osh3 in gray. (D) Structure comparison of apo and PI(4)P-bound ORP3 ORDs. (E) Structure comparison of the PI(4)P-bound ORP3 ORD (this study) and the apo ORP3 ORD (PDB id: 7CYZ).

features of PI(4)P recognition observed in this study is consistent with the recent structural studies of the ORP3 ORD [25]. The key residues interacting with the PI(4)P head group in human ORP3 and yeast Osh3 are strictly conserved with almost identical conformations (Fig 3B). However, the acyl groups of PI(4)P display different conformations in the binding pockets. Many rotatable bonds in the acyl chains and non-specific hydrophobic interactions within the large tunnel seems to allow conformational flexibility of the acyl groups. The PI(4)P used in this structural study has the acyl chains of C8. Ligand modeling into the ORP3 ORD suggests that PI(4)P with various lengths of acyl chains up to C20 could be accommodated in the binding pocket (Fig 3C). The first acryl chain is accommodated in the deep hydrophobic pocket. The second acyl chain is located in the pocket beneath the lid. All the reported structures of the PI(4)P-bound ORDs in a closed conformation displayed a conserved geometry of PI(4)P-recognition for Osh3 [7], Osh4 [6], Osh6 [8] and ORP3, suggesting strict conservation of PI(4)P-binding in the OSBP gene family.

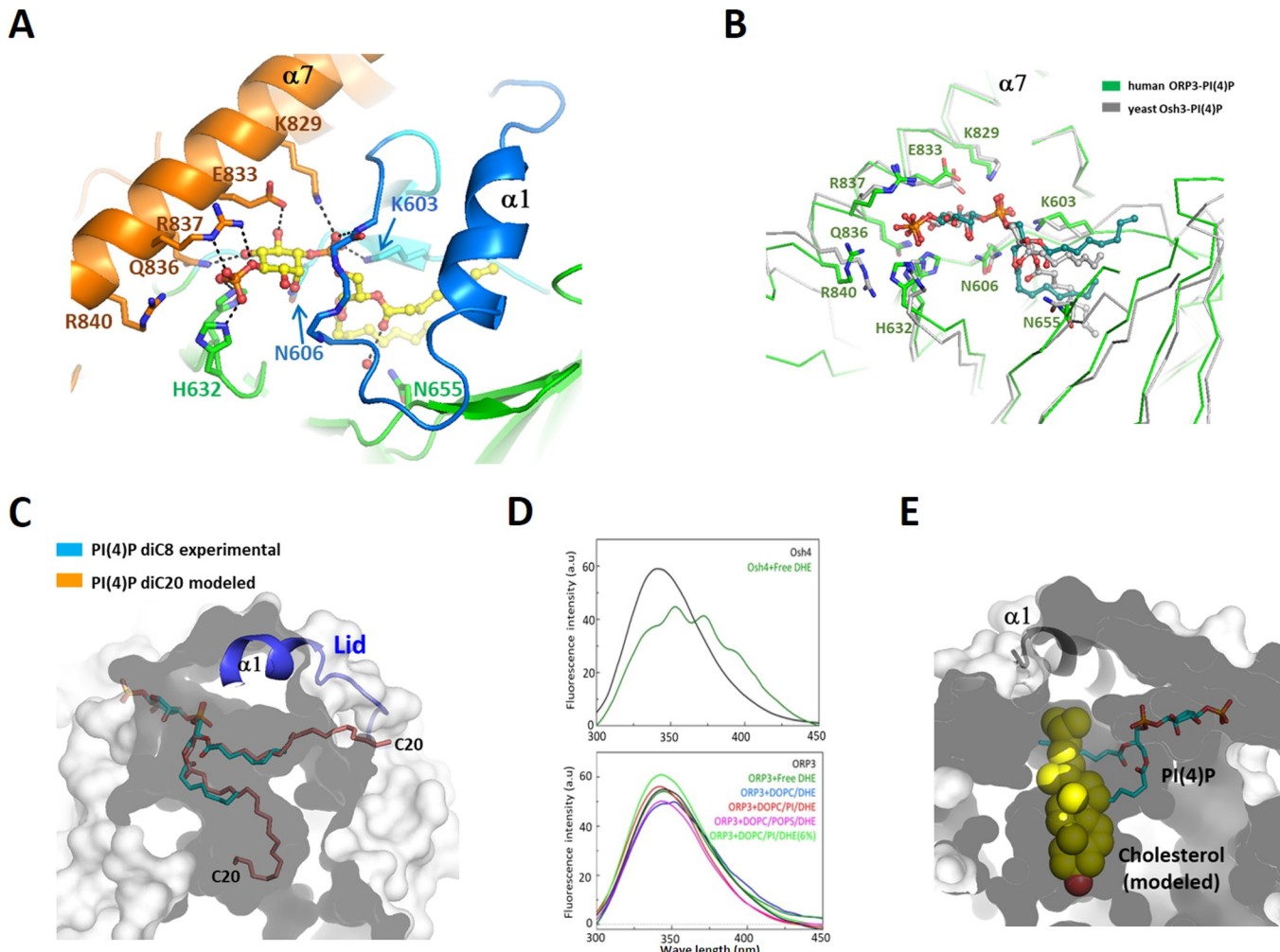

**Fig 3. PI(4)P-binding to the ORP3 ORD.** (A) Binding of PI(4)P diC8 in the ORP3 ORD. The residues interacting with the PI(4)P head group are shown in stick models. Hydrogen bonds with PI(4)P are shown in dotted lines. (B) Structure comparison of the PI(4)P-binding sites of human ORP3 and yeast Osh3. (C) Modeling suggested that PI(4)P diC20 can be accommodated in the binding pocket of ORP3. The acryl chains of PI(4)P diC8 were extended to diC20 and were manually fitted into the pocket. The shape of the ligand-binding pocket is shown as a transparent surface. The surface representation was sliced to visualize the interior of the ORP3 molecule. The outer and inner surfaces of ORP3 were shown in light grey and dark grey, respectively. (D) Dehydroergosterol (DHE) binding of Osh4 and ORP3. Fluorescence emission spectra (excitation wavelength $\lambda_{ex}$ = 285 nm) of the ORD domains incubated with the various liposomes containing DHE. DOPC/DHE (98:2 mol/mol), DOPC/PI/DHE (97.5:0.5:2), or DOPC/POPS/DHE (88:10:2). (E) Cholesterol cannot fit into the hydrophobic tunnel of the ORP3 ORD. The PI(4)P in the binding pocket is shown in a stick model. The cholesterol molecule was modeled into the pocket by superposing the structures of ORP3 and cholesterol-bound ORP1 (PDB id: 5ZM5). The cholesterol atoms clashing with the wall of the hydrophobic pocket are shown in light yellow.

## ORP3 does not bind sterol due to its narrow hydrophobic pocket

OSBP1 binds and transports cholesterol in exchange of PI(4)P between organellar membranes [26], and the several ORPs such as ORP1 [12], ORP2 [13], Osh1 [27], and Osh4 [24], were structurally confirmed to bind sterol as a secondary ligand. To examine the sterol-binding property of ORP3, we performed sterol extraction assay by incubating purified proteins with the liposomes containing fluorescent dehydroergosterol (DHE), a natural ergosterol analogue. Yeast Osh4 extracted DHE efficiently as monitored by a FRET signal between the tryptophan residues of protein and a bound DHE. However, the ORP3 ORD did not extract DHE from the DOPC liposomes composed of several different lipid compositions (Fig 3D). The lack of

sterol binding of ORP3 is consistent with the previous studies [16]. To obtain a structural insight on the inability of ORP3 to bind sterol, we modeled a cholesterol molecule into the hydrophobic pocket of the ORP3 ORD based on the structure of ORP1-cholesterol complex [12]. Sterol binds to the hydrophobic tunnel of several ORPs in a head-down orientation and the 3-hydroxyl group of the sterol is hydrogen bonded to the polar residue on the tunnel bottom. The cholesterol modeling suggested that the tunnel of ORP3 is too narrow to accommodate sterols and clashed with the tetracyclic rings of cholesterol. (Fig 3E). Even though the total cavity volume of the ORP3 ORD is 2.6 times bigger than the volume of cholesterol (622 Å$^3$), the shape of the binding pocket is not compatible for sterol binding. Considering the high sequence conservation of the ORP3 ORD with ORP6 and ORP7, the subfamily III ORPs in human seem to have no sterol-binding properties.

## The essential function of the ORDs in PI(4)P-binding is shared by all ORPs in yeast and human

Human ORP3 localizes to the PM-ER contact site by its N-terminal PH domains and FFAT motif and plays a role in phosphoinositide metabolism at the membrane contact sites [16, 28]. Since yeast Osh homologs share an essential function involving phosphoinositide binding of the ORD [7], it is important to investigate whether the function is also shared by the ORPs of different species. First, we examined the cellular localization of various human ORP3 constructs in yeast CBY926 cells (Fig 4A). The cells transformed with various GFP-ORP3 constructs were grown at 30°C. The full length ORP3 (FL) showed mild cytosolic distribution and punctate localization at the PM (Fig 4B). This is consistent with the localization of the ORP3 at the ER-PM contact sites previously reported [28]. The PM targeting of ORP3 is mediated by the PI(4)P or PI(4,5)P$_2$ binding of the N-terminal PH domain [18]. Consistent with the previous studies, we observed that the ORP3 PH domain alone (PH) showed strong PM localization in yeast cells. The deletion of PH domain from the full length ORP3 (ΔPH) eliminated the punctate localization at the PM. The ΔPH construct was mostly cytosolic but also displayed some punctate distribution inside the cells, possibly due to the presence of an ER-targeting FFAT motif. The major effect of PH deletion was a loss of cortical ER-targeting due to the loss of a PM-binding domain. PH-ORD and ΔFFAT showed both cytosolic and PM distribution, whereas PH domain alone showed more intense PM targeting. The ORD deletion (ΔORD) showed a punctate localization at the PM similar to full length. Consistently, a mutation of phosphoinositide-binding site (KK60-61EE) in the PH-ORD or FFAT deletion constructs eliminated the PM targeting. The ORDs of ORP3 and OSBP1 localized to the cytosol. These observation confirms that the punctate localization at the ER-PM contact sites is determined by the N-terminal PH domain and FFAT motif in yeast.

The yeast *Saccharomyces cerevisiae* has seven ORP genes, *OSH1-OSH7*. Deletion of all seven ORPs in yeast is lethal [29]. The expression of any single *OSH* gene rescues from this lethality, indicating that all Osh homologs share at least one essential function [29]. To examine the importance of individual domains of human ORP3 for the biological function, we tested whether the expression of various mutation and deletion constructs rescue the growth defects of a yeast strain in which all seven *OSH*s have either been deleted, or are present as a temperature-sensitive allele (*oshΔ*[CEN, osh4$^{ts}$]) [22]. As a control, we transformed the wild-type strain (CBY1) with the identical plasmids, and checked the cell growth at 30°C (Fig 4C). Most of the ORP3 and OSBP1 constructs grew well. However, the constructs containing functional PH domain and ORD (PH-ORD and ΔFFAT) caused mild growth suppression for the wild-type strain. In CBY926 cells, the full-length ORP3 rescued the cell growth at 37°C whereas the empty vector did not (Fig 4C). The PH domain and FFAT motif of human ORP3 was not

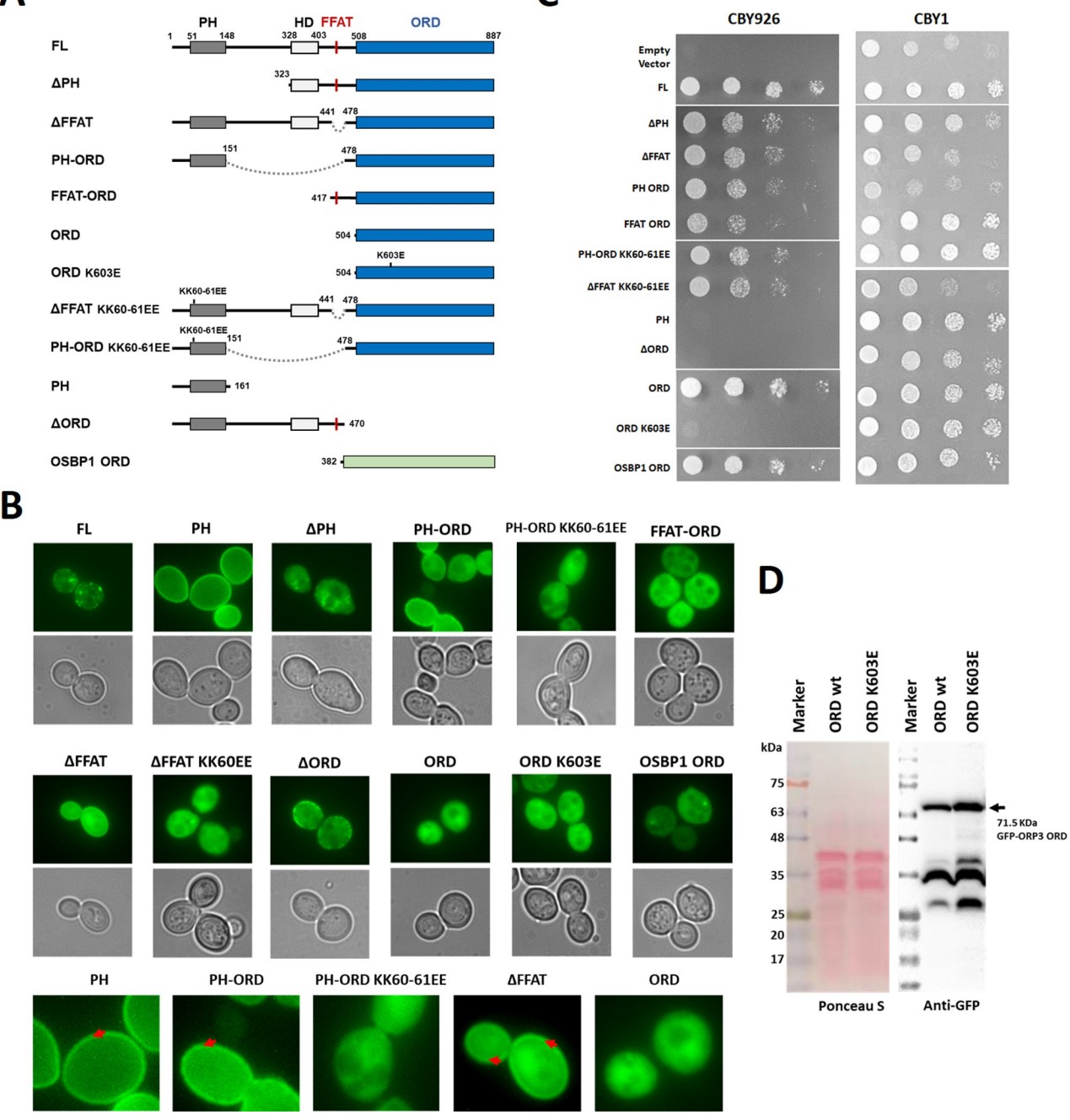

**Fig 4. A shared function of the ORD domain in human and yeast ORPs.** (A) Schematic representation of the constructs used in in yeast complementation and cell imaging. All ORP3 and OSBP1 constructs had an N-terminal GFP fusion in a pRS416MET25 plasmid. The sites of point mutations are indicated on top of the schematic representation. Internal deletions are indicated by dotted lines. (B) Localization of human ORP3 constructs in yeast cells. Lower panels show the zoomed-in images of selected ORP3 constructs. The red arrows indicate PM localization of the GFP-ORP3 constructs. (C) Yeast growth complementation of *oshΔ* [CEN *osh4^ts*] cells co-expressing human ORP3 alleles. The wild type cells (CBY1) were transformed with the same plasmids and were grown at 30°C as a control. (D) The expression levels of GFP-ORP3 wt and K603E mutant in CBY926 cells were examined by Western blot analysis.

essential for the yeast growth (ΔPH and ΔFFAT), even though they have PM and ER targeting function in yeast cell. Consistently, the KK60EE mutations in PH-ORD and ΔFFAT constructs

abolished the PM localization but rescued the yeast growth defect. The ORD alone or the ORDs with the N-terminal targeting domains (PH-ORD or FFAT-ORD) rescued the growth defects at 37˚C. However, the K603E mutation in the ORD that interferes with the PI(4)P-binding was lethal. To confirm the growth defect of the ORD K603E, we performed Western blot analysis to check the expression levels of GFP-ORP3 ORD wild type and K603E (Fig 4D). The K603E mutant expressed 1.5 time higher than the wild type at 30˚C, suggesting that the growth defect of the ORD K603E is caused by a loss of function but not by low expression of the construct. The expression of the ORDs of human ORP3 and OSBP1 rescued the yeast Osh knockout defect, suggesting that the essential function mediated by the ORDs is shared by the ORPs of different species. These data suggest that the minimal requirement for ORP3 function is the presence of the ORD with PI(4)P-binding activity. These results are consistent with the previous studies that the phosphoinositide-binding to the ORDs is essential for the function of ORPs [7, 24].

## Discussion

ORP3 is a cytosolic protein that contains membrane-targeting domains such as PH domain and FFAT motif, displaying 30% of the protein in membrane-bound fraction [28, 30]. Full length ORP3 resides in puncta at the PM, which seems the part of the cortical ER-PM contact sites. The crystal structures of the human ORP3 ORD, together with biochemical and yeast complementation experiments provide a clear picture of the conserved PI(4)P-binding by ORDs and a shared function of the ORP family proteins. This study suggests that the essential function of ORPs is shared between yeast and human and it is mediated by the non-vesicular PI(4)P transport by the ORDs, which does not involve specific protein-protein interactions [31].

Considering the conservation of phosphoinositide-coupled lipid transfer activities of ORPs, ORP3 seems to function by balancing the PI(4)P level of the membranes. Lipid transport by the ORP3 at the ER-PM contact sites seems important for the integrated regulation of phosphoinositide metabolism within the PM [32]. ORP3 was known to lower PI(4)P but not PI(4,5)$P_2$ or PI(3,4,5)$P_3$ levels and it does not influence the level of cholesterol and PS [16]. Osh3, a close ORP3 homolog in yeast, regulates the PI(4)P level at the PM by affecting Sac1 phosphatase at the membrane contact sites [33]. ORP3 does not bind sterol and the secondary ligand for ORP3 was not known, though phosphatidylcholine might be transported by ORP3 in exchange of PI(4)P [18]. Some ORPs such as ORP1 and ORP2, were reported to bind PI(4,5)$P_2$ by adapting an open conformation of the lid and the different orientation of the inositol head group [12, 13]. However, ORP3 displays the conserved mode of PI(4)P-binding and does not seem to bind phsphoinositides other than PI(4)P. The PH domain and FFAT motif allow the targeting of ORP3 to the ER-PM contact sites and the ORD transports PI(4)P by uptake or release of the lipid from the two membranes. Sequential and structural conservation of residues involved in the recognition of PI(4)P suggested that the major role of the subfamily III ORPs including ORP3, ORP6, and ORP7 might be regulation of PI(4)P level in the cells. In conclusion, this study provides structural understanding of human ORP3 ORD on the ligand recognition and the shared function of ORPs in PI(4)P transport.

## Supporting information

**S1 File.**
(PDF)

**S2 File.**
(PDF)

## Acknowledgments

We appreciate C. Beh for providing the yeast CBY1 and CBY926 strains.

## Author Contributions

**Conceptualization:** Young Jun Im.

**Data curation:** Young Jun Im.

**Formal analysis:** Junsen Tong, Young Jun Im.

**Funding acquisition:** Young Jun Im.

**Investigation:** Junsen Tong, Lingchen Tan, Young Jun Im.

**Methodology:** Junsen Tong, Lingchen Tan.

**Project administration:** Young Jun Im.

**Resources:** Young Jun Im.

**Supervision:** Young Jun Im.

**Validation:** Young Jun Im.

**Writing – original draft:** Young Jun Im.

**Writing – review & editing:** Young Jun Im.

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
