## [Decision Letter · Decision Letter 0]

4 Dec 2020

PONE-D-20-34681

Structure of human ORP3 ORD reveals conservation of a key function and ligand specificity in OSBP-related proteins

PLOS ONE

Dear Dr. Im,

Thank you for submitting your manuscript to PLOS ONE. Two expert reviewers provided constructive critiques of the manuscript and made excellent suggestions for improvements. Based on their recommendations, and after careful consideration, we feel that your manuscript has merit but does not fully meet PLOS ONE’s publication criteria as it currently stands. Therefore, we invite you to submit a revised version of the manuscript that addresses the points raised during the review process.

(1) Please address all the suggested changes as described by each reviewer below, and please provide in a separate document an explanation of revisions made.

(2) For Fig.4C, please include wild-type controls for all plasmid transformations. It is presumed that all the constructs will either rescue the osh∆ osh4-1 growth defect or not. Some of these constructs could actually cause dominant growth defects, which would completely change the interpretation of these results. 

(3) Fig.4B, representative cell images are provided but no quantification is provided. How many cells were counted? What percentage of these cells exhibited punctate or cytoplasmic localization? Are the differences statistically significant? 

We look forward to receiving your revised manuscript.

Kind regards,

Christopher Beh, PhD

Academic Editor

PLOS ONE

Journal Requirements:

https://www.cell.com/structure/fulltext/S0969-2126(13)00165-2

The text that needs to be addressed involves the first three subsections of the Results section

In your revision ensure you cite all your sources (including your own works), and quote or rephrase any duplicated text outside the methods section. Further consideration is dependent on these concerns being addressed.

Reviewers' comments:

Reviewer's Responses to Questions

**Comments to the Author**

1. Is the manuscript technically sound, and do the data support the conclusions?

Reviewer #1: Yes

Reviewer #2: Partly

2. Has the statistical analysis been performed appropriately and rigorously? 

Reviewer #1: Yes

Reviewer #2: I Don't Know

3. Have the authors made all data underlying the findings in their manuscript fully available?

Reviewer #1: Yes

Reviewer #2: Yes

4. Is the manuscript presented in an intelligible fashion and written in standard English?

Reviewer #1: Yes

Reviewer #2: Yes

5. Review Comments to the Author

Reviewer #1: This work reports the structure of ORP3-ORD. This is an important contribution to the study of ORPs. There are only some minor concerns.

1. The structures of ORP3-ORD, alone and in complex with PI4P, have been reported (BBRC 529 (2020) 1005-1010). Please compare your structures with the reported ones, and show the structural differences.

2. In addition, PI4P diC8 was used for crystallization trial (both yours and the reported complex), while the physiological PI4P molecules have longer acyl chains (C18/C20). Have you test the binding/extraction of BRAIN PI4P (and PIP2) by ORP3-ORD. Some sort of modelling would help.

3. Page 4, line 53-54, please add ORP2. Also please add the following reference for ORP5/8: PMID: 28970484. References 12 and 13 are for ORP1 and ORP2.

Reviewer #2: The manuscript “Structure of human ORP3 ORD reveals conservation of a key function and ligand specificity in OSBP-related proteins” by Tong et al. presents a high-resolution structural model for human ORP3 and functional heterologous expression data that indicate an essential ORP3 function (and that other ORP family members as well, by inference) is PI4P dependent. The study does not identify the essential function. Nonetheless, the manuscript contributes useful, novel structure-function data to the ORP field in general. One particular strength of the manuscript is its comparative analysis of human and yeast ORP structures and structure-function analysis of human ORP3 in yeast, creating a not so common knowledge bridge between these experimental organisms.

Specific Remarks:

1. A key conclusion of the study is that PI4P binding to the ORP3 ORD is necessary for an essential function in yeast, based on the observation that the ORD K603E allele does not rescue the growth defect of osh delta osh4ts cells at restrictive temperature. Are wild-type ORD and ORD K603E expressed to the same level? If not, the conclusion cannot be made. Also, it is not clear in text or figure legends whether the constructs used in Fig. 4C are the same exact constructs as used in Fig. 4B.

2. Also in regard to Fig. 4B, the authors describe localization of ORP3 and its varying domains in the text (lines 263-276). The images presented do not support all the statements made in the text. For example, line 269, the authors claim that deletion of the PH domain (delta PH) eliminated the punctate localization at the PM. Yet, the image clearly shows punctate localization with the delta PH construct, albeit slightly more diffuse than with wild-type. Further on (line 270), the authors state that PH-ORD and delta FFAT show a membrane localization similar to that of the PH domain. I disagree; there is no similarity evident among the images presented.

3. The Materials and Methods lack information, as follows.

a. Line 82. How was the secondary structure prediction made?

b. Line 85 and 87. The authors report cell growth conditions in Kelvin rather than degree Celsius.

Report in Celsius for ease of reading and keep to one scale throughout the manuscript.

c. Line 92. Define the components of lysis buffer.

d. Line 93. Was the elution with one buffer or two? If one, what was the pH of the buffer with

imidazole added?

e. Line 94. How was the eluate concentrated?

f. Line 95. How much thrombin was used?

g. Line 96. What was the wash and eluent for ion-exchange chromatography?

h. Line 98. How was ORP3 ORD concentrated?

i. Throughout the Materials and Methods, provide the source (supplier) of the reagents used,

especially less common reagents. Same for software applications.

j. Line 105. Typo. Phosphoinositide.

k. Line 145. What type if site-directed mutagenesis?

l. Line 147. CBY926 genotype given as Osh1-7 delta: CEN osh4ts. This conveys, to some readers, that CEN osh4ts is

linked to Osh1-7 delta. Recommend osh delta [CEN, osh4ts] or [pRS416MET25(osh4ts)]. Also, maintain consistency

of nomenclature. Here, the strain is described as Osh1-7 delta; elsewhere in the manuscript it is described as osh

deleta.

m. Line 150. What was the dilution series? 10-fold dilutions? How long were plates incubated? What medium was used?

n. Line 153. Is the pRS416 listed here the same as pRS416MET25 in line 145?

o. Line 155. Again, what medium was used?

p. Line 155. Delete 1X. PBS is 1X unless specified differently.

q. Line 156 and elsewhere. “selective medium”, rather than “selection media”

r. Line 158. What was the exposure time (or range of times) for the image acquisition. Was any imaging processing

performed? If yes, describe.

4. Lines 175-177. The authors reference Figs. 1C and 1D. It appears the text is best supported by Fig. 1B. The purpose

of Fig. 1C is unclear. Fig. 1D is not necessary, but does provide useful data to readers.

5. Line 284. The authors state that the PH domain and FFAT motif of human ORP3 was not essential for yeast growth.

This is true because some yeast growth was observed when these constructs were expressed. However, the authors

should state that only a partial rescue of the growth defect was observed, indicating that this domain and motif serve

some function in yeast.

6. Line 286. The authors refer to the K604E mutation, but Fig. 4 (all panels) indicate the mutation is K603E. Clarify.

7. Line 288. “defect” not “defeat”. Also Line 68.

8. Throughout. “these data” not “this data”

9. Line 457. “Top view of Fig. 2A.” This is Fig. 2A. Do the authors mean to reference another figure?

10. Line 471. The legend for Fig. 3D needs to be more descriptive.

11. Figure4C. PH-ORDKK60EE and DeltaFFAT-KK60EE not described in corresponding text. Describe data in figure.

6. PLOS authors have the option to publish the peer review history of their article (what does this mean?). If published, this will include your full peer review and any attached files.

Reviewer #1: No

Reviewer #2: No

---

## [Author Response · Author response to Decision Letter 0]

6 Jan 2021

We have included point-by-point description to the editor’s and reviewer’s comments in a separate word file.

---

## [Decision Letter · Decision Letter 1]

22 Jan 2021

PONE-D-20-34681R1

Structure of human ORP3 ORD reveals conservation of a key function and ligand specificity in OSBP-related proteins

PLOS ONE

Dear Dr. Im,

Thank you for submitting your manuscript to PLOS ONE. After careful consideration, we feel that it has merit but does not fully meet PLOS ONE’s publication criteria as it currently stands because several concerns were not addressed. Therefore, we invite you to submit a revised version of the manuscript that addresses the points raised during the review process.

1. As was previously requested, there seems to be confusion about the necessity of wild-type transformation controls. The growth of CBY926 cells at 30˚C is not equivalent to wild type (there are obviously many additional genetic deletions involved). In addition, the expression of ORP3 does not have to cause lethality but can cause significant growth defects when expressed in wild-type yeast. Considering previous mistaken presumptions about the genetic nature of such ORP mutations, correctly providing proper controls is necessary for the clarity of such genetic experiments.

2. Reviewer 2 had reasonable concerns that were also not fully addressed. Please see the specific comments below.

Given that all the requested changes are relatively minor and easily finished, it is reasonable that their completion is requisite for paper acceptance.

We look forward to receiving your revised manuscript.

Kind regards,

Christopher Beh, PhD

Academic Editor

PLOS ONE

Reviewers' comments:

Reviewer's Responses to Questions

**Comments to the Author**

1. If the authors have adequately addressed your comments raised in a previous round of review and you feel that this manuscript is now acceptable for publication, you may indicate that here to bypass the “Comments to the Author” section, enter your conflict of interest statement in the “Confidential to Editor” section, and submit your "Accept" recommendation.

Reviewer #1: All comments have been addressed

Reviewer #2: (No Response)

2. Is the manuscript technically sound, and do the data support the conclusions?

Reviewer #1: Yes

Reviewer #2: Partly

3. Has the statistical analysis been performed appropriately and rigorously? 

Reviewer #1: Yes

Reviewer #2: I Don't Know

4. Have the authors made all data underlying the findings in their manuscript fully available?

Reviewer #1: Yes

Reviewer #2: Yes

5. Is the manuscript presented in an intelligible fashion and written in standard English?

Reviewer #1: Yes

Reviewer #2: Yes

6. Review Comments to the Author

Reviewer #1: My concerns have been addressed. This work is a great contribution to the study of this important family of proteins.

Reviewer #2: The authors have satisfactorily addressed the concerns expressed in my previous review, with two exceptions as noted below.

1. The authors have not satisfactorily addressed whether the levels of ORD and ORD K603E are the same. The authors, to support a key assertion in the manuscript, need to show a Western blot, rather than rely on fluorescence microscopy data because a several-fold difference in expression may be undetectable by non-quantitative fluorescence microscopy.

2. The authors continue to over-interpret the fluorescence microscopy data in Fig 4B. In regard to the delta PH construct, the authors describe a punctate distribution "which seems non-cortical ER localization." This assignment of localization appears to be inferred by the presence of the FFAT motif rather than by any data within the figure. Recommend deleting "seems non-cortical ER localization."

With respect to PH-ORD and delta FFAT, the authors assert the localization is both cytosolic and PM, though less PM than with the PH domain alone. This could be true, but it is impossible to distinguish PM from cytosolic localization in the PH-ORD and delta FFAT images provided. The authors need to supply more convincing micrographs or retreat from this conclusion.

7. PLOS authors have the option to publish the peer review history of their article (what does this mean?). If published, this will include your full peer review and any attached files.

Reviewer #1: No

Reviewer #2: No

---

## [Author Response · Author response to Decision Letter 1]

2 Mar 2021

The response to reviewers is attached as a separate word file.

---

## [Editor Report · Decision Letter 2]

5 Mar 2021

Structure of human ORP3 ORD reveals conservation of a key function and ligand specificity in OSBP-related proteins

PONE-D-20-34681R2

Dear Dr. Im,

Thank you for addressing all reviewer concerns. We are pleased to inform you that your manuscript has been judged scientifically suitable for publication and will be formally accepted for publication once it meets all outstanding technical requirements.

Kind regards,

Christopher Beh, PhD

Academic Editor

PLOS ONE
---

## [Editor Report · Acceptance letter]

5 Apr 2021

PONE-D-20-34681R2 

 Structure of human ORP3 ORD reveals conservation of a key function and ligand specificity in OSBP-related proteins 

Dear Dr. Im:

I'm pleased to inform you that your manuscript has been deemed suitable for publication in PLOS ONE. Congratulations! Your manuscript is now with our production department. 

Kind regards, 

on behalf of

Dr. Christopher Beh 

Academic Editor

PLOS ONE